# Robust Multi-bit Text Watermark with LLM-based Paraphrasers

Xiaojun Xu [1]   Jinghan Jia [2] [*]   Yuanshun Yao [1]   Yang Liu [3] [*]   Hang Li [1]

## Abstract

We propose an imperceptible multi-bit text watermark embedded by paraphrasing with LLMs. We fine-tune a pair of LLM paraphrasers that are designed to behave differently so that their paraphrasing difference reflected in the text semantics can be identified by a trained decoder. To embed our multi-bit watermark, we use two paraphrasers alternatively to encode the pre-defined binary code at the sentence level. Then we use a text classifier as the decoder to decode each bit of the watermark. Through extensive experiments, we show that our watermarks can achieve over 99.99% detection AUC with small (1.1B) text paraphrasers while keeping the semantic information of the original sentence. More importantly, our pipeline is robust under word substitution and sentence paraphrasing perturbations and generalizes well to out-of-distributional data. We also show the stealthiness of our watermark with LLM-based evaluation. We open-source the code: https://github.com/xiaojunxu/multi-bit-text-watermark.

## 1. Introduction

Text watermark aims to encode some imperceptible signal into a piece of text so that people are able to decode the signal from the text (Liu et al., 2024). It can be useful in various applications such as copyright protection and hidden message communication. With the development of Large Language Models (LLMs), there is also a growing need to track misinformation spread by LLMs using text watermark injected to model outputs (Kirchenbauer et al., 2023).

We study the methodology of injecting a multi-bit watermark message into a piece of text by paraphrasing. The watermarked text will keep the semantic meaning of the original text after paraphrasing. Another paired decoder will be used to decode the message from the watermarked text. Unlike lexical-based watermarks which inject watermarks by synonym substitutions, the paraphrasing-based method has a larger action space for watermark injection and also is more robust under perturbations. However, there are also challenges in designing paraphrasing-based watermarks, as it is unclear on how to properly inject imperceptible but detectable watermark signal while keeping the text quality and original semantic meaning.

In this work, we propose a paraphrasing-based watermark by simultaneously fine-tuning an LLM-based paraphraser as the encoder and train a LM-based text classifier as the decoder. The pipeline is shown in Figure 1. In the encoding stage, we will paraphrase the input text conditioned on a user-chosen key to generate the watermarked text. In the decoding stage, we will extract the code from the input text with the decoder and compare with the previously chosen key to see if it is watermarked by the user.

The key to produce a high-quality text watermark in our method is to train a good encoder-decoder pair. For the decoder, we can train it with standard classification loss so that it can better classify between "bit-0 texts" and "bit-1 texts". For the encoder, we would like to fine-tune it so that its generated text can be better classified by the decoder. Inspired by (Xu et al., 2024), we show that we can use the decoder as a reward model to evaluate how well the paraphrased text generated by the encoder can be correctly classified. Thus, we can use PPO-based RL techniques to finetune the encoder so that the injected watermark can be better decoded. We adopt a co-training framework so that the encoder and decoder are alternatively updated during the training process.

Through experiments, we show that our experiments can achieve a very high watermark detection performance while maintaining the paraphrasing fidelity. We achieve over 95% bit accuracy and over 0.99 detection AUC, both outperforming existing methods significantly. In addition, we can apply a simple repetition-based strategy and im-

---
[*]Work done while at ByteDance.   [1]ByteDance Research [2]Michigan State University [3]University of California, Santa Cruz. Correspondence to: Xiaojun Xu <xiaojun.xu@bytedance.com>.

prove the detection AUC to over 0.9999. In addition, our method also shows a good robustness under word substitution and sentence paraphrasing perturbations. We also evaluate our methods over out-of-distributional (OOD) data and observe that our model can achieve over 0.99 AUC for most of the OOD tasks. All these results show the effectiveness and robustness of our watermark.

The rest of the paper is organized as follows. We will first introduce the preliminary knowledge of the work in Section 2. Then we introduce our paraphrasing-based watermark methodology in Section 3. We will show the experiment results in Section 4. Finally, we discuss the related work in Section 5 and conclude the work in Section 6.

## 2. Preliminary

### 2.1. Goal of Multi-bit Text Watermark

The goal of the work is to inject a multi-bit watermark message into a piece of text by paraphrasing. Formally speaking, in the watermark injection stage, we are given an original text $x^o$ and a watermark message $M \in \{0,1\}^\infty$. We will inject watermark by generating a new watermarked text with a encoder $x^w = E(x^o, M)$. To extract the watermark, we will use a watermark decoder $M' = D(x^w)$ to decode the injected watermark. We hope that the decoded bits should match the prefix of the designed watermark message, i.e., $M' = M[: \text{len}(M')]$. Note that this is a vary-length watermark, where the length of watermark message is dependent on the length of text - the longer the text is, the more information we can encode in the watermarked text. This is contrary to the fix-length text watermark (e.g. (Zhang et al., 2024b)), where the watermark code is a fixed length for any given input text. The length of $M'$ depend on different watermark designs, and we will introduce them in Section 3.1.

We have the following requirements on the paraphrased text:

- Fidelity: The watermarked text should not change the meaning of the original text. The similarity $sim(x^o, x^w)$ should be high.

- Accuracy: The watermark decoder should accurately decode the watermark message. The error rate $|M' - M[: \text{len}(M')]|_0$ should be low.

- Robustness: The watermark message should still exist after the watermarked text undergoes some perturbation. Let $M'_{pert} = D(\text{pert}(x^w))$ denote decoded message from perturbed watermarked text. We hope

that the error rate after perturbation $|M'_{pert} - M[: \text{len}(M'_{pert})]|_0$ should be low.

- Stealthiness: The watermark should not be easily detected by human eyes. We evaluate it with the criteria that human cannot easily detect the watermarks in the text. Formally speaking, let $M'_h = D_{human}(x^w)$ be the human guess on the watermark code. We hope that $|M'_h - M[: \text{len}(M'_h)]|_0$ should be high, i.e. human guess on the watermark code has a high error rate.

### 2.2. Background: PPO

Proximal Policy Optimization (PPO) (Schulman et al., 2017) is a standard way to optimize a language model towards a high reward calculated by some pre-defined reward functions $r(x) \in \mathbb{R}$, where $x$ is the input text (i.e. a sequence of tokens). Let $\pi(x_t|x_{<t})$ denote the probability of generating token $x_t$ given the context, and $\pi(\cdot|x_{<t})$ denote the overall probability vector. We use $\pi_\theta$ to denote the model to train and $\pi_{ref}$ to denote a reference model. People will first estimate an "advantage" at each step $A_t(x)$ given the final reward $r(x)$, which approximates how each token contributes to the final reward. There are different choices of how to estimate the advantage. We use the Generalized Advantage Estimation (GAE) (Jaques et al., 2019; Zheng et al., 2023) with critic models, which we omit the details here. Having the advantage $A_t(x)$ at each step, the PPO algorithm will optimize the input $x$ by minimizing the following loss:

$$\ell_{PPO}(\theta; x) = \sum_t \Big( - \mathbb{E}_t \big[ \frac{\pi_\theta(x_t|x_{<t})}{\pi_{ref}(x_t|x_{<t})} A_t(x) \big]$$
$$+ \lambda_k \text{KL}(\pi_\theta(\cdot|x_{<t}), \pi_{ref}(\cdot|x_{<t})) \Big) \quad (1)$$

where the first term is to maximize the expected advantage on each token, and the second term is to regularize the model to not drastically change from the reference model.

## 3. Methodology

### 3.1. Overview

We illustrate the high-level pipeline of our watermark in Figure 1. Our core idea is to inject the watermark into a piece of text by paraphrasing the text to include the imperceptible watermark signal, which can be later decoded by a text classifier. To encode a watermark message into a piece of text, we will apply a LLM-based paraphraser conditioned on one watermark bit (0 or 1). The watermark bit is initialized as the first bit of the watermark message, and updated to later bits during the token-by-token generation

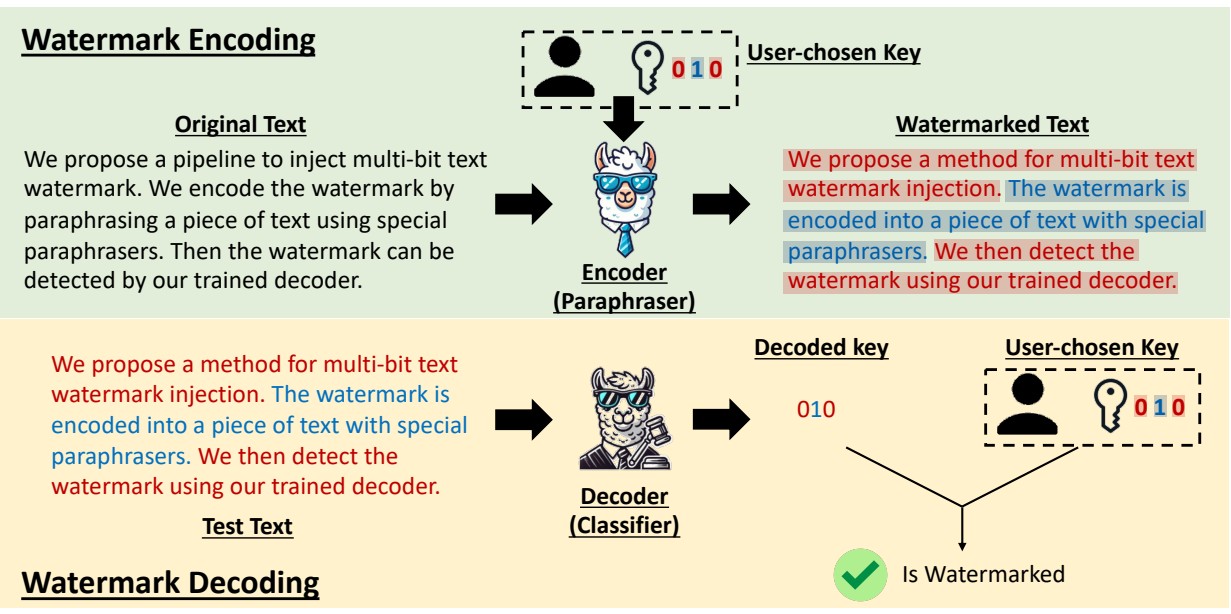

*Figure 1.* The overview of our watermark pipeline. During encoding, we use an encoder to parapharse the input text given a user-chosen key. During decoding, we extract the bits from the text using the decoder.

process. Different segments in the generated text will correspond to different bits in the message code. To decode the watermark message from a piece of watermarked text, we will divide the text into multiple segments, and then apply the LM-based classifier to determine the watermark bit for each segment. The concatenated message is the decoded watermark message.

**Text Segmentor** Note that both processes require a mechanism to divide a text into segments, so that we can assign one bit to each segment of the text to inject multi-bit watermark code. We use a "text segmentor" $\mathcal{S}$ to do the segmentation, which will operate in two different modes during encoding and decoding. During encoding, it will take the current generated text and output a boolean value $\mathcal{S}(x|\text{mode}=E) \in \{0,1\}$ to determine whether the next token will belong to a new segment. During decoding, it will take a piece of text $x$ as input and segment it into a list of segments $\mathcal{S}(x|\text{mode}=D) = [\tilde{x}_1, \tilde{x}_2, \ldots]$. In this work, we choose to do the segmentation on the sentence-level, i.e. every sentence in the text is a segment. We view it as a simple yet robust choice, as word-level injection/deletion will not change the segmentation, and paraphrasing will also keep the sentence order in most cases.

### 3.2. Encoder: LLM-based Paraphraser

The encoder $E$ aims to paraphrase the input text based on a given watermark code and get $x^w = E(x^o, M)$ based

on LLMs. Our design of the encoder is to have two LLM-based paraphrasers $(\theta_0, \theta_1)$ and use them alternatively in the token-by-token generation process, which is based on the current watermark code determined by the sentence segmentor. Formally speaking, let $x_t^w = f(x^o, x_{<t}^w; \theta_i)$ denote the process of generating the next token when paraphrasing the input $x^o$ parametrized by $\theta_i$. The encoding algorithm is shown in Alg. 1. We track the current watermark $bit$, and the next token is generated with the corresponding paraphraser $\theta_{bit}$. After each generation step, we check whether the next token will be in a new segment by calculating $\mathcal{S}(x^w; \text{mode}=E)$. If the new segment starts, we will update $bit$ to be the next bit in the watermark message.

### 3.3. Decoder: LLM-based Text Classifier

The decoder $D$ will decode the watermark code from a piece of text and get $M' = D(x^w) \in \{0,1\}^*$. We use $g(x; \theta_d) \in \{0,1\}$ to denote a binary classifier on a text with parameters $\theta_d$, and use $g_p(x; \theta_d) \in (0,1)$ to denote the predicted probability of class-1. The decoding algorithm is shown in Alg. 2. We will segment the input text into multiple segments $\mathcal{S}(x; \text{mode}=D)$, then apply the classifier to each segment to calculate the decoded watermark.

### 3.4. Co-training Framework

The training framework is inspired by (Xu et al., 2024), which shows that the text classifier can be viewed as a "re-

**Algorithm 1** Watermark Encoding Algorithm $x^w = E(x^o, M; \mathcal{S}, \theta_0, \theta_1)$.

**Require:** Input text $x^o$; Watermark code $M$; Text segmentor $\mathcal{S}$; Parameters for two paraphrasers $\theta_0$ and $\theta_1$.
**Ensure:** Watermarked text $x^w$
1: $x^w \leftarrow [\,]$
2: $i \leftarrow 0$    /* index of current watermark bit */
3: **while** $x^w[-1] \neq \langle\text{EOS}\rangle$ **do**
4:    $bit \leftarrow M[i]$
5:    $x^w$.append($f(x^o, x^w; \theta_{bit})$)
6:    /* Switch to the next bit if the current segmentation ends. */
7:    **if** $\mathcal{S}(x^w; \text{mode}=E) = 1$ **then**
8:        $i \leftarrow i + 1$
9:    **end if**
10: **end while**
11: **return** $x^w$

---

**Algorithm 2** Watermark Decoding Algorithm $M' = D(x^w; \mathcal{S}, \theta_d)$.

**Require:** Input text $x^w$; Text segmentor $\mathcal{S}$; Parameters for the text classifier $\theta_d$.
**Ensure:** Decoded watermark $M'$
1: $M' \leftarrow [\,]$
2: **for** $\tilde{x}_i \in \mathcal{S}(x^w; \text{mode}=D)$ **do**
3:    $M'$.append($g(\tilde{x}_i; \theta_d)$)
4: **end for**
5: **return** $M'$

---

ward model" to finetune LLMs with PPO, and that the text classifier and the LLM can be trained alternatively. In our work, we will alternate between two goals: optimizing the decoder ($\theta_d$) and optimizing the paraphrasers ($\theta_0$ and $\theta_1$). The goal of the decoder is to accurately classify each bit of the original watermark code $M$. We use the cross entropy loss to optimize the decoder:

$$\ell_D(\theta_d; x^w, M) = \sum_{i=1}^{|D(x^w)|} \Big( M[i] \cdot g_s(\tilde{x}_i^w; \theta_d) + (1 - M[i]) \cdot (1 - g_s(\tilde{x}_i^w; \theta_d)) \Big) \quad (2)$$

The goal of the encoder is to generate inputs that can be better recognized by the decoder, while keeping its normal utility (i.e. a good paraphrasing performance). To optimize the encoder, we utilize the idea of PPO that a LLM can be fine-tuned with RL-based techniques with respect to a reward model. Here, the decoder is used to calculate the "reward" of how the output of encoder can be successfully decoded as the original watermark code. Specifically,

given original text $x^o$, watermark code $M$ and the watermarked text $x^w = E(x^o, M)$, the watermark reward $r_w$ is calculated by:

$$r_w(x^w, M) = \sum_{i=1}^{\text{len}(D(x^w))} \mathbb{1}\{D(x^w)[i] = M[i]\} \quad (3)$$

In addition, we will also calculate a similarity reward $r_s(x^w, x^o)$ with a text similarity model. The overall reward is a weighted sum of the two rewards:

$$r(x^w, x^o, M) = \lambda_w \cdot r_w(x^w, M) + \lambda_s \cdot r_s(x^w, x^o) \quad (4)$$

Having the reward, we will use the PPO algorithm to update the parameters $(\theta_0, \theta_1)$. One change in our PPO loss is that our $x^w$ is generated by two models $\theta_0$ and $\theta_1$, so each model only needs to update on the inputs that are generated by each model. The formal PPO loss for encoder, assuming we have calculated the advantage $A_t(x_w, x_o, M)$ (which we will abbreviate as $A_t$ without ambiguity), is as follows:

$$\begin{aligned}\ell_E(\theta_0, \theta_1) = &\sum_t \mathbb{1}\{x_t \sim \pi_{\theta_0}(\cdot|x_{<t})\} \cdot \Big( -\mathbb{E}_t\big[\frac{\pi_{\theta_0}(x_t|x_{<t})}{\pi_{ref}(x_t|x_{<t})} A_t\big] \\ &+ \lambda_k \text{KL}(\pi_{\theta_0}(\cdot|x_{<t}), \pi_{ref}(\cdot|x_{<t}))\Big) \\ &+ \sum_t \mathbb{1}\{x_t \sim \pi_{\theta_1}(\cdot|x_{<t})\} \cdot \Big( -\mathbb{E}_t\big[\frac{\pi_{\theta_1}(x_t|x_{<t})}{\pi_{ref}(x_t|x_{<t})} A_t\big] \\ &+ \lambda_k \text{KL}(\pi_{\theta_1}(\cdot|x_{<t}), \pi_{ref}(\cdot|x_{<t}))\Big)\end{aligned}$$
$$(5)$$

where the information of whether $x_t$ is generated by $\theta_0$ or $\theta_1$ is recorded during the generation stage.

The algorithm is shown in Algorithm 3. We will have a dataset consisting of original texts $x_o$. In each training step, we randomly sample a watermark key $M$. Then we calculate the watermarked text $x_w$ with the current encoder $(\theta_0, \theta_1)$ and the advantage function with the current decoder $\theta_d$. Finally, we update the encoder and decoder with the respective losses.

**Initialization** In practice, we observe that the training performance heavily depends on the model initialization. This is expected, as the encoder and decoder rely on each other to do the update and therefore requires a good initialization - the update of $(\theta_0, \theta_1)$ needs the reward provided by $\theta_d$, and the update of $\theta_d$ needs the samples generated by $(\theta_0, \theta_1)$. In our implementation, we will first initialize $(\theta_0, \theta_1)$ with supervised finetuning (SFT) loss on a paraphrasing dataset $\mathcal{D}_{SFT} = \{(x_o^{SFT}, x_{para}^{SFT})\}$. We will simultaneously finetune the two models $\theta_0$ and

---

**Algorithm 3** Training Algorithm of the Encoder and the Decoder.

**Require:** Dataset $\mathcal{D}$; Initialized parameters $\theta_0, \theta_1, \theta_d$; Text Segmentor $\mathcal{S}$
 1:
**Ensure:** Trained parameters $\theta_0, \theta_1, \theta_d$
 2:
 3: **for all** $x_o \in \mathcal{D}$ **do**
 4:     $M \sim \{0, 1\}^\infty$
 5:     $x_w \leftarrow E(x^o, M; \mathcal{S}, \theta_0, \theta_1)$
 6:     Calculate the advantage function $A_t(x_w, x_o, M)$ with the reward function in Equation 4.
 7:     Update $\theta_d$ with decoder loss $\ell_D(\theta_d; x^w, M)$ in Equation 2.
 8:     Update $\theta_0, \theta_1$ with the encoder loss $\ell_E(\theta_0, \theta_1; A_t)$ in Equation 5.
 9: **end for**
10: **return** $M'$

---

$\theta_1$ on the paraphrasing dataset and hope that they both have a small loss, but they also have a difference in their behaviour (measured by JS divergence), with the loss $\ell_{init}(\theta_0, \theta_1; x_o^{SFT}, x_{para}^{SFT})$ (denoted as $\ell_{init}(\theta_0, \theta_1)$ for simplicity) as follows:

$$
\begin{aligned}
\ell_{init}(\theta_0, \theta_1;) = {} & \ell_{SFT}(\theta_0; x_o^{SFT}, x_{para}^{SFT}) \\
& + \ell_{SFT}(\theta_1; x_o^{SFT}, x_{para}^{SFT}) \\
& - \lambda_{JS} \cdot \text{JS}(\pi_{\theta_0}(x_{para}^{SFT}|x_o^{SFT}), \pi_{\theta_1}(x_{para}^{SFT}|x_o^{SFT}))
\end{aligned}
\tag{6}
$$

After the paraphrasers are finetuned, we will generate watermarked texts $x^w$ with randomly sampled watermark code $M$, and initialize the decoder by optimizing $\ell_D(\theta_d; x^w, M)$ in Equation 2.

## 4. Experiments

### 4.1. Setting

**Model and Training Settings** We use a relatively small TinyLlama-1.1b model architecture (Zhang et al., 2024a) for $\theta_0$, $\theta_1$ and $\theta_d$, as we observe that small models can already achieve a good performance in paraphrasing and watermarking. We show the experiments with larger Llama-2-7b models in Appendix C. The detailed prompt used by the pararphrasers are shown in Figure 3 in Appendix A. The encoder and decoder are trained and evaluated on the C4 RealNewsLike dataset (Raffel et al., 2020), processed using standard settings in (Kirchenbauer et al., 2023; Xu et al., 2024; Lau et al., 2024). Without specification, we will use texts with 128 tokens for training and evaluation.

We fine-tune the model for 10,000 steps with batch size of 4. We use $\lambda_w = 0.1$, $\lambda_s = 1.0$ and $\lambda_k = 0.02$ as the coefficients. In the initialization stage, we will generate the paraphrased data $x_{para}^{SFT}$ with Pegasus paraphraser (Zhang et al., 2020), and use $\lambda_{JS} = 1.0$ for the intialization loss.

**Metric** We evaluate three types of metrics of a text watermark. The first type is the bit-wise accuracy, which evaluates how good the multi-bit watermark code is extracted. This includes the bit-wise accuracy (Bit Acc) of the decoded watermark and the number of total bits injected in the text (Bit Num). The second type is the text-wise accuracy, which evaluates how well we can tell the watermarked text apart from other non-watermarked text. We will evaluate the decoder on both watermarked and non-watermarked texts, and calculate the area under ROC curve (AUC) and true positive rate under 1%, 0.01% false positve (TPR@FPR=1%, TPR@FPR=0.01%). For the fidelity, we calculate the similarity with the `all-mpnet-base-v2`[1] model following the setting in (Lau et al., 2024).

**Baselines** We evaluate various baseline methods with different design ideas:

- RemarkLLM (Zhang et al., 2024b). The idea is to use a fixed-length multi-bit watermark key and train a Transformer-based paraphraser with a watermark detector. The paraphraser is trained with Gumbel reparametrization techniques to minimize the decoding error. We use the T5-based paraphraser in their original setting and evaluate both the 4-bit version and 8-bit version of the watermarking model.

- KGW (Kirchenbauer et al., 2023) and KTH (Kuditipudi et al., 2023). They are LLM-based watermarks aiming to inject watermark to LLM-generated texts by altering the token sampling strategy during the generation stage of a LLM. Note that their methods are not directly comparable with ours, as they are not designed to watermark non-LLM-generated text. For comparison, we adapt them to watermark any text with two variant, zero-bit and multi-bit. In the zero-bit variant, we directly apply KGW or KTH to a LLM-based (1.1B) paraphraser, which is then used to paraphrase the given text to inject watermarks. This is a zero-bit watermark as the detector can only tell whether a text is watermarked or not, but no other information will be carried in the watermark. In the multi-bit variant, we will apply KGW or KTH to two LLM-based paraphrasers. Then we use them as $\theta_0$ and

---

[1]`https://huggingface.co/`
`sentence-transformers/all-mpnet-base-v2`

*Table 1.* The performance of our watermark compared with baseline methods. The RemarkLLM method uses the T5 (Raffel et al., 2020) model following their original settings. Other methods use TinyLlama-1.1B (Zhang et al., 2024a) as the paraphraser. The bit-wise accuracy is marked as "-" if the method does not support multi-bit watermark code.

| Method | Bit-wise Accuracy | | Text-wise Accuracy | | | Fidelity |
|---|---|---|---|---|---|---|
| | Bit Acc | Bit Num | AUC | TPR@FPR=1% | TPR@FPR=0.01% | Similarity |
| RemarkLLM (4bit) | 0.7663 | 4.0 | 0.7861 | 0.0% | 0.0% | 0.8096 |
| RemarkLLM (8bit) | 0.6953 | **8.0** | 0.8023 | 3.7% | 0.0% | 0.7793 |
| KGW (zero-bit) | - | - | 0.8652 | 25.9% | 18.1% | 0.7745 |
| KGW (multi-bit) | 0.6381 | 4.46 | 0.8327 | 22.9% | 6.3% | 0.8123 |
| KTH (zero-bit) | - | - | 0.8919 | 61.4% | 46.6% | 0.8200 |
| KTH (multi-bit) | 0.6129 | 4.26 | 0.6775 | 10.9% | 2.3% | 0.8176 |
| Waterfall($\kappa = 0.5$) | - | - | 0.7787 | 14.0% | 3.8% | 0.8499 |
| Waterfall($\kappa = 1$) | - | - | 0.9392 | 62.4% | 35.5% | 0.8423 |
| Ours | **0.9563** | 5.57 | **0.9981** | **98.0%** | **78.0%** | **0.8739** |

$\theta_1$ in our approach and paraphrase one text based on a watermark code. This allows the multi-bit information to be carried in the watermark.

- Waterfall (Lau et al., 2024). They prompt a pretrained Llama model as the paraphraser and will change the sampling stage in order to inject the watermark signal. Their extracted watermark code is a permutation, which does not support bit-wise comparison. We evaluate the watermark strength at $\kappa = 0.5$ and $\kappa = 1$. Note that in their original paper, they use a strong watermark up to $\kappa = 8$. However, in our evaluation, we observe that even $\kappa = 2$ will affect the paraphrasing performance significantly for the 1.1B small model. Therefore, we use a relatively small $\kappa$ in the evaluation.

Note that we did not compare with some well-known text watermark as they are already covered in previous works. We did not compare with AWT (Abdelnabi & Fritz, 2021) as RemarkLLM shows a better performance in their paper. We did not compare with Robust Multi-bit (Yoo et al., 2023) and NLW (Qiang et al., 2023) as Waterfall shows a better performance in their paper. There are also many works (e.g. (Christ et al., 2024; Zhao et al., 2023)) that focus on LLM watermarks, but we only choose the representative ones (KGW and KTH).

### 4.2. Performance

We show the watermark performance in Table 1. We can observe that our method achieves a better performance than existing methods on both bit-wise accuracy and text-wise accuracy. Our method also has high information density, with approximately one bit per 23 tokens (128/5.57). In

addition, we also observe a higher similarity score compared to baseline methods. This might be surprising at first glance. We owe it to the reason that we add a similarity reward during the PPO process, so that the model is fine-tuned to achieve a good paraphrasing performance.

**Multiple run** In paraphrasing-based watermark, we can run the paraphraser multiple times and return the result with best watermark detection rate. This method is adopted in previous methods (Zhang et al., 2024b; Lau et al., 2024). In this section, we evaluate how different methods improve with multiple runs of the paraphraser. The results are shown in Figure 2. We can observe that our methods can scale to over 0.99 bit accuracy and 0.9999 detection AUC with five repeats of the paraphraser. Since we use a 1.1B small model which can be run in parallel efficiently, we view it as a good tradeoff to repeat five times and achieve a better watermark performance. Other methods also get a performance boost with more repeats, but there is still a clear performance gap.

**Example and Analysis on Stealthiness** We show several examples of the watermarked text and their original version in Table 6 in Appendix B. The sentences of class 0 and class 1 are marked with blue and green respectively. All the sentences are correctly classified by the decoder. From our observation, it is difficult to tell a significant difference between the two classes of sentences, confirming the stealthiness of our watermark.

To further validate the stealthiness of our watermark, we prompt GPT with in-context learning to see if it can tell the difference between the two classes of sentences. Specifically, we provide GPT with ten class-0 and ten class-1 sentences, and ask it to classify which class a new sentence

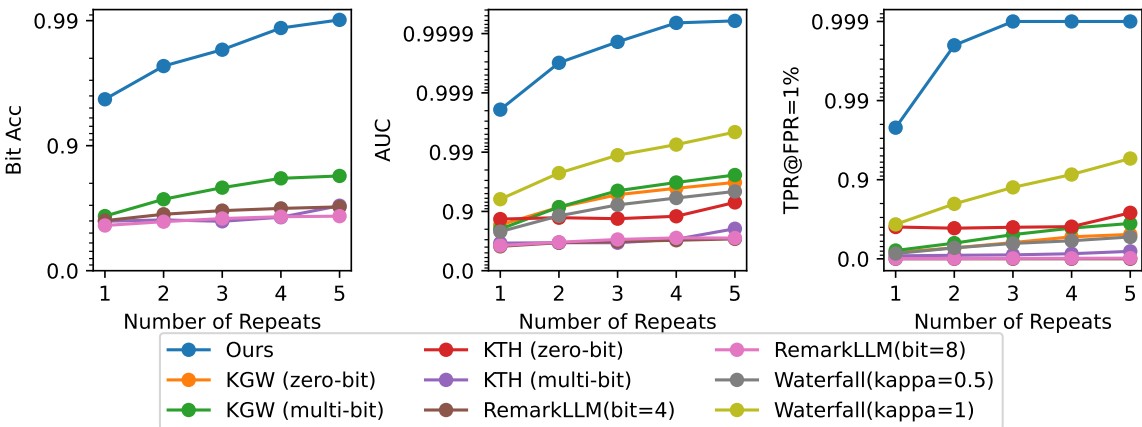

*Figure 2.* The detection performance of our watermark compared with baseline methods with multiple repeats of the paraphraser. Note that some methods do not support multi-bit watermark code, so they do not have a curve of bit accuracy in the left figure.

*Table 2.* The performance of our watermark compared with baseline methods under word substitution attack.

| Method | Substitute ratio 5% | | | Substitute ratio 10% | | | Substitute ratio 20% | | |
|---|---|---|---|---|---|---|---|---|---|
| | bitacc | AUC | TPR@1% | bitacc | AUC | TPR@1% | bitacc | AUC | TPR@1% |
| RemarkLLM (4bit) | 0.6118 | 0.6215 | 0.0% | 0.6315 | 0.6441 | 0.0% | 0.6488 | 0.6624 | 0.0 |
| RemarkLLM (8bit) | 0.5685 | 0.6281 | 0.6% | 0.5783 | 0.6445 | 1.0% | 0.5921 | 0.6665 | 0.8% |
| KGW (zero-bit) | - | 0.8458 | 21.4% | - | 0.8353 | 16.5% | - | 0.7779 | 7.0% |
| KGW (multi-bit) | 0.6208 | 0.8052 | 20.9% | 0.6134 | 0.7914 | 18.9% | 0.5840 | 0.7471 | 12.8% |
| KTH (zero-bit) | - | 0.8718 | 56.5% | - | 0.8541 | 51.8% | - | 0.8128 | 41.5% |
| KTH (multi-bit) | 0.6018 | 0.6574 | 9.0% | 0.5955 | 0.6504 | 8.0% | 0.5610 | 0.6120 | 5.1% |
| Waterfall($\kappa = 0.5$) | - | 0.7578 | 12.5% | - | 0.7344 | 9.1% | - | 0.6893 | 5.3% |
| Waterfall($\kappa = 1$) | - | 0.9250 | 54.1% | - | 0.9096 | 28.9% | - | 0.8558 | 25.6% |
| Ours | 0.9382 | 0.9945 | 93.5% | 0.9193 | 0.9871 | 86.4% | 0.8605 | 0.9469 | 51.6% |
| Ours(advt) | **0.9459** | **0.9958** | **94.1%** | **0.9352** | **0.9936** | **91.6%** | **0.9138** | **0.9853** | **78.7%** |

belongs to. The detailed prompt is shown in Figure 4 in Appendix A. We evaluate 1,000 class-0 and 1,000 class-1 sentences, and the accuracy is 57.0%, which is close to the performance of random guess (50.0%). Thus, we conclude that our watermark is stealthy and it is difficult to tell a difference between the two classes of sentences.

## 4.3. Robustness

In this section, we study the robustness of our watermark. The evaluation pipeline follows the standard protocol - we first generate a watermarked text, then modify the text with text-level perturbations, and finally test whether we can still detect the watermark in the text. We will evaluate word substitution and sentence paraphrasing, which are two most popular perturbations on texts. In addition to our benign-trained model, we also evaluate the adversarially trained model (denoted as Ours-AdvT), which has the knowledge of perturbation during training and will use the perturbed text when training the decoder.

**Word Substitution** For paraphrasing attack, we will randomly substitute {5%, 10%, 20%} tokens in the text with another randomly chosen token (uniformly sampled from the vocabulary). We show the results in Table 2. The adversarial training model uses 10% of word substitution during the training process. We can observe that our original model can already outperform all the baselines when perturbed with word substitutions. With the knowledge of perturbation during the training process, we can further improve the performance and achieve over 0.99 detection AUC even when 10% of the tokens are randomly substituted.

**Sentence Paraphrasing** For sentence paraphrasing, we consider three types. Following (Lau et al., 2024), we will translate the sentence to Spanish and then back to English with a Llama2-7B model, denoted as "Translate". We will also directly prompt a Llama2-7B model to paraphrase the sentence, denoted as "LlamaPara". The detailed prompts used to do the translation and paraphrasing are shown in

*Table 3.* The performance of our watermark compared with baseline methods under sentence paraphrasing attack.

| Method | Translate | | | LlamaPara | | | PegasusPara | | |
|---|---|---|---|---|---|---|---|---|---|
| | bitacc | AUC | TPR@1% | bitacc | AUC | TPR@1% | bitacc | AUC | TPR@1% |
| RemarkLLM (4bit) | 0.6885 | 0.7142 | 0.0% | 0.7063 | 0.7311 | 0.0% | 0.7033 | 0.7248 | 0.0% |
| RemarkLLM (8bit) | 0.6124 | 0.6904 | 1.4% | 0.6023 | 0.6751 | 1.5% | 0.6018 | 0.6687 | 1.2% |
| KGW (zero-bit) | - | 0.4872 | 0.2% | - | 0.4872 | 0.2% | - | 0.4900 | 0.0% |
| KGW (multi-bit) | 0.4997 | 0.5829 | 1.6% | 0.4765 | 0.5383 | 1.5% | 0.4817 | 0.5654 | 1.5% |
| KTH (zero-bit) | - | 0.8600 | 30.6% | - | 0.8559 | 32.0% | - | 0.8618 | 43.7% |
| KTH (multi-bit) | 0.4923 | 0.4990 | 0.8% | 0.4952 | 0.4957 | 1.7% | 0.4949 | 0.5025 | 1.3% |
| Waterfall($\kappa = 0.5$) | - | 0.6041 | 4.0% | - | 0.5833 | 1.9% | - | 0.5981 | 5.0% |
| Waterfall($\kappa = 1$) | - | 0.7432 | 11.8% | - | 0.6519 | 3.1% | - | 0.7283 | 13.2% |
| Ours | 0.8206 | 0.9310 | 67.4% | 0.7137 | 0.8649 | **43.9%** | 0.7388 | 0.8616 | **53.7%** |
| Ours(advt) | **0.9003** | **0.9709** | **78.1%** | **0.8487** | **0.9239** | 36.8% | **0.8648** | **0.9546** | 45.7% |

*Table 4.* The performance of our watermark, trained on the C4 dataset, when evaluated on texts collected in other tasks.

| Dataset | Bit-wise Accuracy | | Text-wise Accuracy | | | Fidelity |
|---|---|---|---|---|---|---|
| | Bit Acc | Bit Num | AUC | TPR@FPR=1% | TPR@FPR=0.01% | Similarity |
| HH | 0.9582 | 5.856 | 0.9991 | 97.9% | 92.1% | 0.8823 |
| PKU | 0.9613 | 5.325 | 0.9959 | 96.7% | 1.8% | 0.8923 |
| Reward | 0.9572 | 5.684 | 0.9962 | 96.7% | 51.4% | 0.8711 |
| UltraF | 0.9519 | 6.234 | 0.9931 | 94.5% | 55.7% | 0.8830 |
| FineWeb | 0.9461 | 6.066 | 0.9880 | 93.3% | 19.3% | 0.8463 |
| Pile | 0.9140 | 6.026 | 0.9713 | 83.8% | 36.1% | 0.8430 |

Figure 5 and 6 in Appendix A. In addition, following (Xu et al., 2024), we also paraphrase the sentence with the Pegasus (Zhang et al., 2020) paraphraser, denoted as "PegasusPara".

The results are shown in Table 3. We observe that all these text watermarking methods suffer from a significant performance drop under paraphrasing attacks. We owe it to the reason that the text watermarks aim to preserve the text meaning and inject watermarks with other signals (e.g. wording choices or stylish changes), while these signals will be easily broken by another paraphrasing process. As an extreme example, one may paraphrase the watermarked text into its original un-watermarked version (because the watermarking process requires that both texts should have the same semantic meaning), and it is impossible to detect the watermark from the text after perturbation (i.e. the original text). Nevertheless, it is still possible to preserve part of the watermark signal under mild paraphrasing, such as translation. We can observe that our method can outperform baselines on all the paraphrasing tasks, and can be further improved with adversarial training.

### 4.4. Out-of-Distributional Tasks

As our pipeline relies on a data-driven training process, we would like to evaluate how it performs on potential out-of-distribution data. In this section, we will evaluate our model, previously trained on the C4 dataset, on various other datasets, including Anthropic HH-RLHF (HH) (Bai et al., 2022), Synthetic instruction[2](Instruct), PKU SafeRLHF (PKU) (Ji et al., 2024), Reward[3], UltraFeedback(UltraF) (Cui et al., 2024), FineWeb (Penedo et al., 2024) and Pile uncopyrighted(Pile)[4] datasets. Among the datasets, HH, Instruct, PKU, Reward and UltraF are QA datasets for alignment and we use their answers as the original texts. FineWeb is a dataset consisting of articles from the Internet. Pile is a dataset consisting of cleaned texts from different sources.

The performance of our model is shown in Table 4. We can observe that our model can generally achieve a good

---

[2]https://huggingface.co/datasets/Dahoas/synthetic-instruct-gptj-pairwise

[3]https://huggingface.co/datasets/yitingxie/rlhf-reward-datasets

[4]https://huggingface.co/datasets/monology/pile-uncopyrighted

*Table 5.* Performance of our watermark when varying regularization coefficients $\lambda_s$ and $\lambda_k$.

| Coefficients | Bit Acc | AUC | Similarity |
|---|---|---|---|
| $\lambda_s = 5.0, \lambda_k = 0.02$ | 0.7606 | 0.9028 | **0.9728** |
| $\lambda_s = 2.0, \lambda_k = 0.02$ | 0.9525 | 0.9967 | 0.8961 |
| $\lambda_s = 1.0, \lambda_k = 0.02$ | 0.9563 | 0.9981 | 0.8739 |
| $\lambda_s = 0.5, \lambda_k = 0.02$ | 0.9679 | **0.9988** | 0.8515 |
| $\lambda_s = 0.2, \lambda_k = 0.02$ | **0.9722** | 0.9987 | 0.8283 |
| $\lambda_s = 1.0, \lambda_k = 0.1$ | 0.9036 | 0.9739 | **0.8878** |
| $\lambda_s = 1.0, \lambda_k = 0.05$ | 0.9284 | 0.9849 | 0.8840 |
| $\lambda_s = 1.0, \lambda_k = 0.02$ | 0.9563 | 0.9981 | 0.8739 |
| $\lambda_s = 1.0, \lambda_k = 0.01$ | 0.9799 | **0.9991** | 0.8529 |
| $\lambda_s = 1.0, \lambda_k = 0.005$ | **0.9828** | **0.9991** | 0.8489 |

performance on different datasets, indicating its good generalization capability. We do observe a relatively weak performance on the Pile task, which we view as a result of the frequent structural texts (e.g. XML languages) in the dataset. Nevertheless, we emphasize that we can always include a new data domain in the training process, so that they become "in-domain" and can achieve a higher performance.

### 4.5. Impact of $\lambda_s$ and $\lambda_k$

As discussed in Section 3.4, we use $\lambda_s$ to control the similarity reward regularization and $\lambda_k$ to control the KL divergence regularization in the process of paraphraser training. In this subsection, we study how these coefficients impact the final training performance. Specifically, we vary the coefficients from their original choice $\lambda_s = 1.0, \lambda_k = 0.02$ and show the resulted detection performance and sentence similarity in Table 5. We can observe that $\lambda_s$ and $\lambda_k$ indeed control the trade-off between detectability and fidelity - when we increase the coefficient, fidelity will be improved but the detectability will be decreased. Nevertheless, the performance is good for both aspects in most coefficient selections. We view our choice in the main experiments to have a moderate tradeoff between fidelity and detectability.

## 5. Related Works

**Text Watermarks** People have been studying text watermarks for a long time in order to protect copyrights (Liu et al., 2024). Early works on text watermarks focus on synonym substitution or other direct changes in the text. (Topkara et al., 2006) proposes to add watermarks to a text by replacing the most ambiguous words with synonyms in a text. (Xiang et al., 2018) investigated the frequency of synonym words so that more bits can be injected with the frequency information. (Munyer et al., 2024) considers the Word2Vec embedding (Mikolov, 2013) in the synonym substitution so that more information can be injected. (Yoo et al., 2023) extracts invariant features from the text to substitute synonyms so that the watermark can be more robust under different perturbations. More recently, people have studied how to directly inject watermark by paraphrasing the text. (Abdelnabi & Fritz, 2021) proposes a LSTM-based pipeline to paraphrase a text and inject a fixed number of watermark bits. (Zhang et al., 2024b) improves the work by using Transformer-based pipeline and proposing to use Gumbel softmax for token selection conditioned on the watermark code. (Lau et al., 2024) proposes to use an LLM-based paraphraser and inject watermarks in the permutations of n-gram information in the text.

**LLM Output Watermarks** Besides text watermarking, there is also a line of research which studies the injection of watermarks into LLMs, so that the output texts of a LLM can be later detected. (Kirchenbauer et al., 2023) first proposes to watermark an LLM. They will increase the logits of certain random tokens, which are generated based on n-gram information. They then perform a statistical test on the text to determine whether the token appearance frequency is from the watermarked LLM. Follow-up works (Hou et al., 2023; Liu et al., 2023) will generate the random tokens based on semantic meaning rather than n-gram information, which makes the watermark robust against paraphrasing attacks. (Kuditipudi et al., 2023) adds perturbation during the sampling phase after the logits are generated, so that there is no distributional change on the output text. (Gu et al., 2023) proposes to distill a watermarked model into a new LLM model with changed parameters, so that no special mechanism is required during inference. (Xu et al., 2024) proposes a co-training framework on the watermarked LLM and a watermark detector so that the detector is trained to detect the watermarked text and the LLM is finetuned to get easily detected. Unlike text watermarking, this line of work focuses purely on LLM-generated text.

## 6. Conclusion

In this work, we propose a multi-bit text watermark by paraphrasing a piece of text to inject watermark signals. We show that our pipeline achieves very high detection accuracy with good fidelity and stealthiness. In addition, our method is robust under different attacks. Our method sheds new light on the study of text watermarks.

## Impact Statement

This paper proposes a method to inject a binary watermark code into a piece of text. Our method can help with the problem of LLM-generated text tracking and human text copyright protection. However, it may also be applied in applications such as hidden message convey, where someone encrypts the code into a text in a stealthy way.

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

## A. Prompts Used in the Experiments

We show the detailed prompts used in the experiments as below:

- Figure 3: prompt used in the encoder.

- Figure 4: prompt used to do in-context classification with GPT.

- Figure 5: prompt used to translate a text with Llama-2-7B.

- Figure 6: prompt used to paraphrase a text with Llama-2-7B.

We did not make special efforts to optimize these prompts.

```
Human: Paraphrase the text below.
{Original Text}
Assistant: Paraphrased Text:
```

*Figure 3.* The prompt used to paraphrase the text in the encoder.

```
    I have two classes of text, C1 and C2, which have some intrinsic difference.
I will provide you with lists of texts from bothclasses. Can you help me
classify which class a new text is in? You answer should only contain one word,
[C1] or [C2].
    C1 texts:
    {Class-0 sentences}

    C2 texts:
    {Class-1 sentences}

    New text:
    {The new sentence to classify}

    Please answer C1 or C2.
```

*Figure 4.* The prompt used to performance in-context classification of our watermarked text with GPT.

```
    [[INST]] <<SYS>> Translate the provided piece of text to {language}. Do not
include any other sentences after the response, such as explanations of the
translation.
    <</SYS>>

    {text} [/INST]

    Here is a translated version of the text:
```

*Figure 5.* The prompt used to evaluate the watermark robustness under translation.

## B. Examples of Watermarked Texts

We show the watermarked texts generated by our pipeline in Table 6. Blue and green texts correspond to class-0 and class-1 texts respectively. We view it difficult to tell a difference between the two classes of texts from human eyes.

## C. Experiments on Llama-2-7B Models

We show the results of using Llama-2-7B model as the paraphraser in Table 7. Note that the RemarkLLM method does not support Llama models, so we do not evaluate the method; the Waterfall method on 7B models can support a larger $\kappa$,

*Table 6.* Examples of watermarked texts. Blue and green texts correspond to class-0 and class-1 texts respectively.

| Original Text | Watermarked Text | Similarity |
| --- | --- | --- |
| "When it comes to fantasy sports and betting on NASCAR races, there's nothing wrong with it," Gaughan said. "I wanted to go all in on gambling last year," NASCAR executive Steve O'Donnell said. "We have so many people that are linked to the cars. I think the integrity is a big piece to it," O'Donnell said. Nevada's effective monopoly on sports betting ended last spring, when the Supreme Court ruled the ban should be | "There's nothing wrong with fantasy sports and betting on NASCAR races," Gaughan said. Steve said I wanted to go all in on gambling last year. "We have so many people that are linked to the cars," O'Donnell said. The integrity of the car is a big piece to it because they are linked to it. Nevada's effective monopoly on sports betting ended last spring, as the Supreme Court ruled that the ban should be | 0.9177 |
| President Trump's decision Monday to revive plans to freeze federal employee pay in 2020 and to institute a series of cuts to federal employee retirement programs was met with great consternation from stakeholder groups, although the ideas stand little chance of becoming law. Increasing employee contributions toward federal defined benefit annuity programs by 1 percent per year until those payments reach 50 percent of the total cost. Eliminating cost of living adjustments for FERS retirees, and reducing CSRS cost of living adjustments by 0.5 percent. | President Trump's decision Monday to resume plans to freeze federal employee pay and to cut retirement benefits for federal employees generated consternation from stakeholder groups, despite having little hope of becoming law. The employee contributions to the annuity programs are up by 1 percent a year until they reach five percent of the total cost. There are cost of living adjustments for FERS retirees and cost adjustments for COLA, which are reduced by 0.5 percent. | 0.8947 |
| Bob "Bus Bob" Krause, 59, of Waikiki, an Oahu Transit System bus driver, died at home. He was born in Bremen, Germany. He is survived by parents Hans Krause and Sonja Aiwohi, brother Ralph and sisters Lorraine Kinnamon and Charmaine Moniz. Celebration of life: 2 p.m. Friday at Outrigger Canoe Club Waikiki. Additional celebration of life: 4:30 p.m. on weekend of May 4 and 5 at K | Bob "Bus Bob" Krause, the head driver of the Oahu Transit System, died at home. His parents lived in Germany when he was born. He has surviving relatives, including his mother, sister, and brother. The celebration of life is on Friday at the outrigger canoe club. There is a celebration of life on Friday, May 4 and 5 at K | 0.8743 |
| Occasional diarrhea is a common occurrence. Most people will experience an episode of diarrhea at least once or twice a year that will disappear in a couple of days. Luckily, there are many foods to eat that may help a person reduce the symptoms of diarrhea. There are also some foods to avoid when dealing with a bout of diarrhea, and some additional home care tips to consider. Anyone who is experiencing persistent diarrhea should see a doctor, as a person may become dehydrated over time. | Occasional diarrhea is a common occurrence. People will get sick more often than they used to do. There are many foods to eat that may help a person reduce the symptoms of diarrhea. A lot of people avoid foods when they are dealing with a bout of diarrhea and a few home care ideas to consider are worth checking out. Anyone who is suffering from persistent diarrhea should see a doctor, as a person may become dehydrated over time. | 0.8392 |

```
    [[INST]] <<SYS>> Paraphrase the user provided text while preserving semantic
similarity. Do not include any other sentences in the response, such as explanations
of the paraphrasing. Do not summarize.
    <</SYS>>

    {text} [/INST]

    Here is a paraphrased version of the text:
```

*Figure 6.* The prompt used to evaluate the watermark robustness under Llama paraphrasing.

*Table 7.* The performance of our watermark compared with baseline methods with the Llama-2-7B model.

| Method | Bit-wise Accuracy | | Text-wise Accuracy | | | Fidelity |
|---|---|---|---|---|---|---|
| | Bit Acc | Bit Num | AUC | TPR@FPR=1% | TPR@FPR=0.01% | Similarity |
| KGW (zero-bit) | - | - | 0.8625 | 24.4% | 13.7% | 0.8842 |
| KGW (multi-bit) | 0.6302 | 5.17 | 0.8498 | 15.2% | 8.3% | 0.8986 |
| KTH (zero-bit) | - | - | 0.8735 | 26.5% | 12.5% | **0.9075** |
| KTH (multi-bit) | 0.5756 | 5.075 | 0.7296 | 13.3% | 2.0% | 0.9073 |
| Waterfall($\kappa = 1$) | - | - | 0.7568 | 13.3% | 3.7% | 0.8809 |
| Waterfall($\kappa = 2$) | - | - | 0.9213 | 49.3% | 26.9% | 0.8743 |
| Waterfall($\kappa = 4$) | - | - | 0.9951 | 96.3% | **89.8%** | 0.8350 |
| Ours | 0.9605 | 5.874 | **0.9973** | **97.6%** | 77.6% | 0.8631 |

so we included results of $\kappa = 1, 2, 4$ in the table. We can observe that our model keeps a high performance with the 7B models. We do not see an improvement compared with the 1.1B models, which we guess is because that fine-tuned 1.1B models already have the capability to paraphrase texts, so that a larger model may not help. On the other hand, baseline methods can have a better fidelity with the larger model. The Waterfall methods are able to use larger $\kappa$ to inject strong watermarks, and the strongest $\kappa = 4$ case can achieve a comparable performance with our model, though there would be a drop on the fidelity.

