# OpenReview forum: "Robust Multi-bit Text Watermark with LLM-based Paraphrasers"
_ICML.cc/2025/Conference — ICML 2025 poster_

### Official Review · Reviewer_3TNX · 2025-03-11

**Overall Recommendation:** 3

**Summary:**

The paper introduces methodologies for embedding imperceptible multi-bit text watermarks. The proposed algorithm aims to fine-tune a pair of LLM paraphrasers that are designed to behave differently so that their paraphrasing difference reflected in the text semantics can be identified by a trained decoder. The effectiveness of the proposed method is demonstrated through extensive experiments.

**Claims And Evidence:**

The claims are supported by clear evidence.

**Essential References Not Discussed:**

None

**Experimental Designs Or Analyses:**

Missing ablation study. For example, does the similarity reward $r_s$ (Equation 4) remain critical when using larger models (e.g., Llama-2-7B)?

**Methods And Evaluation Criteria:**

Methods: The co-training framework (encoder-decoder with PPO-based RL) is novel and appropriate for aligning watermark injection and detection. Sentence-level segmentation simplifies multi-bit encoding.

Evaluation: Metrics (bit accuracy, AUC, similarity scores) are well-chosen. Baselines (RemarkLLM, KGW, KTH, Waterfall) cover representative works.

**Other Comments Or Suggestions:**

The first occurrence of an abbreviation should be given in full (e.g., LLM in Abstract).

**Other Strengths And Weaknesses:**

Strengths:

The presentation is clear and easy to follow. The authors explain the proposed algorithm clearly.

The performance on several datasets shows the effectiveness of the proposed algorithm.

Weakness:

Lack of ablation experiments to show the effectiveness of the similarity reward $r_s$ in Equation 4.

Some experimental details are not clearly described. In the experimental metrics, how do you determine a text is watermarked (the extracted multi-bit watermark is consistent with the injected multi-bit watermark? Or most of bits in the extracted multi-bit watermark are correct).

**Questions For Authors:**

If a larger model is used, such as Llama-2-7b, is the similarity reward $r_s$ in Equation 4 still needed?

In the experiment, during the calculation of TPR@FPR=1%, how to determine whether a text is the watermarked text based on the extracted multi-bit watermark information?

**Relation To Broader Scientific Literature:**

The work builds on paraphrasing-based watermarks. It uses PPO-based RL techniques to finetune the encoder so that the injected watermark can be better decoded.

**Theoretical Claims:**

No explicit theoretical proofs are provided.

---

> ### Author Rebuttal · Authors · 2025-03-28
>
> **Ablation study of similarity reward** (Lack of ablation experiments to show the effectiveness of the similarity reward rs in Equation 4. ... If a larger model is used, such as Llama-2-7b, is the similarity reward rs in Equation 4 still needed?)
>
> **Response**: We show the effect of similarity reward $r_s$ with an ablation study on its coefficient $\lambda_s$, as well as another ablation study on $\lambda_k$ which controls the weight of the KL divergence term. The result is shown in the table below. We can observe that $\lambda_s$ and $\lambda_k$ indeed controls the trade-off between detectability and fidelity - when we increase the coefficient, fidelity will be improved but the detectability will be decreased. This shows that the similarity reward $r_s$ is important in training a high-performance paraphraser.
>
> Regarding the second question of whether $r_s$ is still needed for larger models, we argue that it is still essential in order to preserve good paraphrasing performance. This is because if we only train the model with watermark loss, its paraphrasing ability will be decreased during the training process. Therefore, we still need the similarity reward to maintain its performance.
>
> | Task | bitAcc | AUC | Similarity |
> | :---- | ----: | ----: | ----: |
> | $\lambda_s$=5.0 | 0.7606 | 0.9028 | **0.9728** |
> | $\lambda_s$=2.0 | 0.9525 | 0.9967 | 0.8961 |
> | $\lambda_s$=1.0 (original) | 0.9563 | 0.9981 | 0.8739 |
> | $\lambda_s$=0.5 | 0.9678 | **0.9988** | 0.8515 |
> | $\lambda_s$=0.2 | **0.9722** | 0.9987 | 0.8283 |
>
> | Task | bitAcc | AUC | Similarity |
> | :---- | ----: | ----: | ----: |
> | $\lambda_k$=0.1 | 0.9036 | 0.9739 | **0.8878** |
> | $\lambda_k$=0.05 | 0.9284 | 0.9849 | 0.8840 |
> | $\lambda_k$=0.02 (original) | 0.9563 | 0.9981 | 0.8739 |
> | $\lambda_k$=0.01 | 0.9799 | **0.9991** | 0.8529 |
> | $\lambda_k$=0.005 | **0.9828** | **0.9991** | 0.8489 |
>
> **Clarification on metrics** (In the experimental metrics, how do you determine a text is watermarked (the extracted multi-bit watermark is consistent with the injected multi-bit watermark? Or most of bits in the extracted multi-bit watermark are correct).)
>
> **Response**: We are sorry for the confusion here. We will determine whether a text is watermarked by checking the proportion of the extracted watermark bits that match the injected bits. For example, if 90% of the bits match, then the text is watermarked. Note that the threshold (90% in the example) is not a fixed value but will be changed in order to calculate AUC and TPR@FPR=x. We will clarify the metric computation in the revision of our paper.
>
> **Writing** (The first occurrence of an abbreviation should be given in full (e.g., LLM in Abstract).)
>
> **Response**: We thank the reviewer for pointing out the writing issues. We will fix them in the revision of the paper.

---

> > ### Comment · Reviewer_3TNX · 2025-04-06
> >
> > I have read the responses by authrs and have no further questions.

---

### Official Review · Reviewer_ZSdw · 2025-03-13

**Overall Recommendation:** 3

**Summary:**

The paper proposes a method for injecting watermarks to text. The key idea proposed is to use two LLM paraphrasers (one for the '0' bit
and one for the '1' bit). The decoder (classifier) and the paraphrasers are trained together in a co-training setup. The results achieved are impressive, achieving 99.9% AUC and 95% bit accuracy while using relatively small (1.1B) models. The authors also conduct robustness checks by perturbing the text and show that their method outperforms other baselines in the presence of word substitution and sentence paraphrasing attacks.

**Claims And Evidence:**

The key claims on the performance of their framework are well-supported by the experimental evidence provided. Specifically, the authors claim their framework achieves high watermark detection performance (table 1), is robust to word substitution and paraphrasing attacks (tables 2 and 3) and generalizes to out of distribution data (Appendix B).

**Essential References Not Discussed:**

I was not able to find papers that were not cited.

**Experimental Designs Or Analyses:**

Overall, the experimental design and analyses are sound. The authors evaluate on multiple datasets, compare against appropriate baselines, and test different perturbation attacks.
The stealthiness claims are a bit weak, as there is no subjective evaluation performed large scale by human participants. Instead the authors use a GPT based model to check.

**Methods And Evaluation Criteria:**

The evaluation criteria of using AUC and robustness checks make sense for this problem.

**Other Comments Or Suggestions:**

The paper is well written, the claims would be strong if there are ablation studies on length of text, choice of text segmentation and computational analysis.

**Other Strengths And Weaknesses:**

Strengths:
1. The authors propose a novel approach of having LLM phrasers for each bit and training them together in a co-training framework.
2. They show strong empirical results on detection accuracy and robustness on multiple datasets. They also compare their work against relevant baselines to show improvement.
3. The paper is well-written and structured.

Weakness:
1. There are no ablation studies on the choice of sentence segmentation. I understand that it was an assumption, but what is the impact of using a different segmentation method?
2. There are no studies on the length of the text, i.e how is the performance when dealing with small texts vs large documents.
3. There are no analysis on the computational overload compared to other baselines, considering we use two llms.

**Questions For Authors:**

I put my questions in the weakness section above. Specifically, my questions are related to how the performance and robustness of this method is for texts of varying length. Also how is the choice of text segmentation affecting the performance.

**Relation To Broader Scientific Literature:**

The existing literature could be classified to pre-LLM based watermarks and LLM based watermarks. The contributions of the paper fall under LLM based text watermarking. Within this category, there are three key works: Remark LLM that uses LLM paraphrasers, KGW and KTH that add watermarks during token generation, making it unsuitable for non LLM generated text. Although LLM based watermarking was explored before, this paper shows that having two LLM paraphrasers, trained in a co-training setup can perform better than the other methods in terms of performance and robustness check.

**Theoretical Claims:**

The paper does not have any theoretical claims.

---

> ### Author Rebuttal · Authors · 2025-03-28
>
> **Different Segmentation** (what is the impact of using a different segmentation method?)
>
> **Response**:  To compare the performance of different segmentation strategies, we conduct an extra experiment in which we design a "segment-by-token" strategy, where we segment the text every 20 tokens and show the result in the table below. We can observe that the segment-by-token strategy also works well under normal scenario and substitution attacks. However, the performance significantly drops under translation attacks. This is because the token order will be changed after being translated back and forth, so that the segmentation of the perturbed watermarked text ($\mathcal{S}(pert(x^w))$) will be different from that of the watermarked text ($\mathcal{S}(x^w)$). This also explains why we choose our current segmentation strategy - it is generally robust under both word substitution and parapharsing. As discussed in Section 6, we view the investigation of other segmentation strategies as an important future work.
>
> | Task | bitacc | AUC | TPR@1% | Similarity
> | :---- | ----: | ----: | ----: | ----: |
> | Ours, segment-every-20-tokens | 0.9507 | **0.9987** | **98.6%** | 0.8667 |
> | Ours, segment-by-sentence (original) | **0.9563** | 0.9981 | 98.0% | **0.8739** |
>
> | Task | bitacc | AUC | TPR@1% |
> | :---- | ----: | ----: | ----: |
> | Ours, segment-every-20-tokens, under substitution (10%) | **0.9242** | **0.9917** | **90.1%** |
> | Ours, segment-by-sentence (original), under substitution (10%) | 0.9193 | 0.9871 | 86.4% |
> | Ours, segment-every-20-tokens, under translation | 0.6015 | 0.6835 | 1.8% |
> | Ours, segment-by-sentence (original), under translation | **0.8206** | **0.9310** | **67.4%** |
>
> **Different length of text** (There are no studies on the length of the text, i.e how is the performance when dealing with small texts vs large documents.)
>
> **Response**: We thank the reviewer for pointing out the necessity to do ablation studies on input text length. We perform the experiment of varying input text length and show the results in the table below, where "len" refers to the number of tokens in the input text. We can observe that the bit-wise accuracy keeps similar, while the text-wise detection AUC grows as the input length grows. This is expected, as longer text (and thus a longer watermark code) will provide better error tolerance for detection. Interestingly, paraphrasing similarity also grows with longer text. This is probably due to the fact that shorter sentences have less possibilities of being changed, so that the paraphrasers need to do more changes in order to inject watermarks.
>
> | Task | bitAcc | AUC | Similarity |
> | :---- | ----: | ----: | ----: |
> | len=16 | 0.9417 | 0.8133 |  0.7639 |
> | len=32 | 0.9463 | 0.8908 | 0.8064 |
> | len=64 | 0.9521 | 0.9698 |  0.8396 |
> | len=128 (original) | 0.9563 | 0.9981 | 0.8739 |
>
> **Computation Overhead** (There are no analysis on the computational overload compared to other baselines, considering we use two llms.)
>
> **Response**: Our model, similar to other text watermark methods like RemarkLLM and Waterfall, uses a LLM-based paraphraser to inject watermarks into text. The main computation is the time required to run the LLM-based paraphraser, where we use two small models (1.1B). Since we need to do forward pass for two models, the runtime is approximately similar to that of running a 2.2B model. By comparison, the Waterfall approach runs a 13B model. The earlier RemarkLLM work uses the T5 model, which is a smaller 220M model. We hypothesize that such a small model may lead to a relatively lower paraphrasing performance. As shown in Table 1 in our paper, their similarity score is around 0.8 while ours can achieve >0.87. In addition, we show the average time to run one watermark injection process in the table below. Surprisingly, the runtime of our method and RemarkLLM (for which we use their open-source implementation) is roughly the same. We owe it to the reason that current LLM packages, e.g. Huggingface Transformers, have better optimization for more recent models like Llama.
>
> | Method | # Parameters | Average Runtime (sec) |
> | :---- | ----: | ----: |
> | RemarkLLM | 220M (T5) | 1.13 |
> | Waterfall | 13B (Llama) | 7.92 |
> | Ours | 1.1B*2 (TinyLlama) | 1.23 |

---

> > ### Comment · Reviewer_ZSdw · 2025-04-02
> >
> > Thanks for the explanation. I am happy to upgrade my decision to an accept, as this is a good solid methodology paper demonstrating strong performance improvements.

---

> > > ### Author Response · Authors · 2025-04-02
> > >
> > > We are glad that the reviewer appreciates our methodology and performance. Thank you for your thoughtful review and positive feedback!

---

### Official Review · Reviewer_E75M · 2025-03-21

**Overall Recommendation:** 3

**Summary:**

The paper presents a robust multi-bit text watermarking method that leverages LLM-based paraphrasers to embed imperceptible watermark signals into text while maintaining semantic fidelity. The approach involves fine-tuning a pair of paraphrasers designed to generate text variations that encode a predefined binary watermark at the sentence level. A trained LLM-based text classifier is then used as a decoder to retrieve the watermark from the modified text. The method employs a co-training framework using Proximal Policy Optimization (PPO), where the encoder (paraphraser) and decoder (classifier) are trained iteratively to optimize watermark embedding and extraction.

**Claims And Evidence:**

Yes

**Essential References Not Discussed:**

Yes

**Experimental Designs Or Analyses:**

Yes

**Methods And Evaluation Criteria:**

Yes

**Other Comments Or Suggestions:**

See the Weakness

**Other Strengths And Weaknesses:**

Strength
- Unlike traditional watermarking approaches that rely on lexical substitutions, this method leverages paraphrasing, providing a larger action space for embedding watermarks while maintaining naturalness in generated text.
- The proposed watermarking method achieves over 99.99% detection AUC, outperforming existing techniques.

Weakness

- The method's performance is highly reliant on model initialization and carefully chosen hyperparameters (e.g., λw, λs, λk). To what extent do these hyperparameters influence the effectiveness of the approach? This dependency raises concerns regarding the method's reproducibility and its robustness when applied to different datasets or scaled-up models.

- The experiments primarily use relatively small models, such as TinyLlama-1.1B. Could you discuss on why larger models, like Llama-2-7B, show only marginal performance improvements?

- If the watermarking technique becomes publicly available, an attacker could develop an adaptive strategy specifically designed for the watermarking process. How can the proposed watermarking method effectively defend such adaptive attacks?

**Questions For Authors:**

See the Weakness

**Relation To Broader Scientific Literature:**

The paper proposes a Robust Multi-bit Text Watermark.

**Theoretical Claims:**

Yes

---

> ### Author Rebuttal · Authors · 2025-03-28
>
> **Ablation Study** (The method's performance is highly reliant on model initialization and carefully chosen hyperparameters (e.g., λw, λs, λk). To what extent do these hyperparameters influence the effectiveness of the approach?)
>
> **Response**: We agree with the reviewer that the hyperparameter choices are important to the performance of our pipeline. To evaluate their impact, we conduct ablation studies that fix $\lambda_w$ and change the other two coefficients. The results are shown in the tables below. We can observe that $\lambda_s$ and $\lambda_k$ indeed control the trade-off between detectability and fidelity - the larger these coefficients, the better the paraphrasing performance but the worse the watermark detectability. Nevertheless, we argue that in most cases, performance is good for both detectability and fidelity.
>
> | Task | bitAcc | AUC | Similarity |
> | :---- | ----: | ----: | ----: |
> | $\lambda_s$=5.0 | 0.7606 | 0.9028 | **0.9728** |
> | $\lambda_s$=2.0 | 0.9525 | 0.9967 | 0.8961 |
> | $\lambda_s$=1.0 (original) | 0.9563 | 0.9981 | 0.8739 |
> | $\lambda_s$=0.5 | 0.9678 | **0.9988** | 0.8515 |
> | $\lambda_s$=0.2 | **0.9722** | 0.9987 | 0.8283 |
>
> | Task | bitAcc | AUC | Similarity |
> | :---- | ----: | ----: | ----: |
> | $\lambda_k$=0.1 | 0.9036 | 0.9739 | **0.8878** |
> | $\lambda_k$=0.05 | 0.9284 | 0.9849 | 0.8840 |
> | $\lambda_k$=0.02 (original) | 0.9563 | 0.9981 | 0.8739 |
> | $\lambda_k$=0.01 | 0.9799 | **0.9991** | 0.8529 |
> | $\lambda_k$=0.005 | **0.9828** | **0.9991** | 0.8489 |
>
> **Larger Model** (Could you discuss on why larger models, like Llama-2-7B, show only marginal performance improvements?)
>
> **Response**: We owe it to the reason that the paraphrasing task is a relatively easy task so that small models can be fine-tuned to achieve good performance. For example, although the PEGASUS paraphrasing model was proposed in 2020 and has less than 600M parameters, it has good paraphrasing performance and is still widely used in current paraphrasing tasks. With the similarity reward included during our RL process, our 1.1B models are fine-tuned to be good paraphrasers. Therefore, using larger models may only provide marginal improvements.
>
> **Adaptive Attacks** (How can the proposed watermarking method effectively defend such adaptive attacks?)
>
> **Response**: We thank the reviewer for pointing out the possibility of adaptive attacks. We can think of two possibilities of adaptive attacks. First, the adversary can attack the detection model with adversarial ML techniques, which aims to slightly change the text so that the detection model output is greatly changed. For this attack, we argue that the detection model parameters will be kept private to the watermark provider and not available by the adversary. Therefore, this is a black-box adversarial attack on LLM-based detectors which, as far as we can tell, does not have a well recognized attack method that works well. We welcome suggestions on potential attacks against our watermark detector under the black-box setting.
>
> Another possibility of adaptive attack is to hack the text segmentation process. For example, knowing that our watermark is segmented based on sentences, the adversary may try to insert or delete sentences so that the watermark code cannot be recognized. That is to say, suppose the text is assigned watermark code 1010110, the adversary can delete the second sentence to make it 110110, and thus the matching rate between the ground truth and decoded code will be low (only 2 bits are matched). To mitigate this exact problem, we may use the longest common sequence algorithm to calculate the match rate (in this case, 5 bits will be matched). In general, we argue that the segmentation method can also be varied and kept private, thus reducing the problem of being hacked. For example, in the response to Reviewer ZSdw and yNYD, we show that we may also segment the text by every 20 tokens, and also achieve a good watermark performance.

---

### Official Review · Reviewer_yNYD · 2025-03-24

**Overall Recommendation:** 3

**Summary:**

The authors proposed a multi-bit text watermark by paraphrasing a piece of text to inject watermark signals. The watermark consists of an encoder-decoder pair. The encoder is fine-tuned to generate text that is classified by the decoder. The decoder is trained with a classification loss to better classify between bit-0 texts and bit-1 texts.

**Claims And Evidence:**

Yes.

**Essential References Not Discussed:**

No.

**Experimental Designs Or Analyses:**

Yes.

**Methods And Evaluation Criteria:**

Are there any other evaluation metrics to evaluate the performance of the watermarking texts?

**Other Comments Or Suggestions:**

No.

**Other Strengths And Weaknesses:**

Strengths of the paper:
1. The paper is well-written and easy to follow.
2. The problem is of great value to investigate.
3. Overview of the proposed model is provided in the figure.

Weaknesses of the paper:
1. The text segmentor simply consider each sentence in the text as a segment. Are there other better segment strategies?
2. The authors consider similarity etc. as evaluation metrics to evaluate the watermarked texts. Are there other evaluation metrics that can be taken for evaluation purpose?
3. How good is the proposed watermarking in terms of avoiding additional computational burden?
4. The authors are encouraged to make the source code of the proposed model publicly available such that the experimental results are convincing to other researchers.
5. Does the proposed watermark method change the parameter of the original LLM? If yes, how does the change of the parameters affect the performance of the original LLM?

**Questions For Authors:**

Please see the above weaknesses.

**Relation To Broader Scientific Literature:**

Other researchers working on LLMs may be interested in the topic of this paper.

**Theoretical Claims:**

Yes.

---

> ### Author Rebuttal · Authors · 2025-03-28
>
> **Different Segmentor** (The text segmentor simply consider each sentence in the text as a segment. Are there other better segment strategies?)
>
> **Response**: We agree with the reviewer that we use a simple segment strategy which splits text by sentences. Nevertheless, we argue that our current strategy is an effective choice, since the segmentation will be robust under word substitution and paraphrasing. To compare the performance of different segment strategies, we conduct an extra ablation experiment. We segment the text every 20 tokens and show the result in the table below. We can observe that the segment-by-token strategy also works well under normal scenario and substitution attacks. However, the performance significantly drops under translation attacks. This is because the token order will be changed after being translated back and forth, so that the segmentation of the perturbed watermarked text ($\mathcal{S}(pert(x^w))$) will be different from that of the watermarked text ($\mathcal{S}(x^w)$). As discussed in Section 6, we view the investigation of other segment strategies as an important future direction.
>
> | Task | bitacc | AUC | TPR@1% | Similarity |
> | :---- | ----: | ----: | ----: | ----: |
> | Ours, segment-every-20-tokens | 0.9507 | **0.9987** | **98.6%** | 0.8667 |
> | Ours, segment-by-sentence (original) | **0.9563** | 0.9981 | 98.0% | **0.8739** |
>
> | Task | bitacc | AUC | TPR@1% |
> | :---- | ----: | ----: | ----: |
> | Ours, segment-every-20-tokens, under substitution (10%) | **0.9242** | **0.9917** | **90.1%** |
> | Ours, segment-by-sentence (original), under substitution (10%) | 0.9193 | 0.9871 | 86.4% |
> | Ours, segment-every-20-tokens, under translation | 0.6015 | 0.6835 | 1.8% |
> | Ours, segment-by-sentence (original), under translation | **0.8206** | **0.9310** | **67.4%** |
>
> **Other evaluation metrics** (The authors consider similarity etc. as evaluation metrics to evaluate the watermarked texts. Are there other evaluation metrics that can be taken for evaluation purpose?)
>
> **Response**: We thank the reviewer for bringing out the question of metric design. We use three types of metrics to evaluate the watermarked texts - the bit-wise accuracy, the text-wise accuracy and the fidelity. These three metrics are the most commonly used metrics in the related works (e.g. RemarkLLM, Waterfall). We welcome suggestions on other metrics that could be helpful to evaluate the watermark performance.
>
> **Computation** (How good is the proposed watermarking in terms of avoiding additional computational burden?)
>
> **Response**: Our model, similar to other text watermark methods like RemarkLLM and Waterfall, uses a LLM-based paraphraser to inject watermarks into text. The main runtime overhead is the time required to run the LLM-based paraphraser. To reduce computational burden, we use two small models (1.1B) and show that we can achieve good paraphrasing performance. By comparison, the Waterfall approach runs a 13B model. Nevertheless, we notice that the earlier RemarkLLM work uses the T5 model, which is a smaller 220M model. We hypothesize that such a small model may lead to a relatively lower paraphrasing performance. As shown in Table 1 in our paper, their similarity score is around 0.8 while ours can achieve >0.87.
>
> In addition, we show the average time to run one watermark injection process in the table below. Surprisingly, the runtime of our method and RemarkLLM (for which we use their open-sourced implementation) is roughly the same. We owe it to the reason that current LLM packages, e.g. Huggingface Transformers, have better optimization for more recent models like Llama.
>
> | Method | # Parameters | Average Runtime (sec) |
> | :---- | ----: | ----: |
> | RemarkLLM | 220M (T5) | 1.13 |
> | Waterfall | 13B (Llama) | 7.92 |
> | Ours | 1.1B*2 (TinyLlama) | 1.23 |
>
> **Open-sourcing** (The authors are encouraged to make the source code of the proposed model publicly available such that the experimental results are convincing to other researchers.)
>
> **Response**: We thank the reviewer for pointing out the necessity of open-sourcing. We do have an open-source plan and promise to release the code of our work if the paper is accepted.
>
> **Original model Parameters** (Does the proposed watermark method change the parameter of the original LLM?)
>
> **Response**: As a clarification, we will have two different classes of LLMs in our pipeline. First, for the watermark injection, we will have "paraphrasing LLMs" to paraphrase the text to inject the watermark. The parameters of these paraphrasing LLMs are finetuned so that they can be used to inject watermark signals. Second, as a text watermark is often applied to watermark LLM-generated texts, we will apply our watermark method to texts that are generated by a "source LLM". We hypothesize that the reviewer is referring to this source LLM, whose parameters will not be changed in our watermark algorithm.

---

### Decision · Program_Chairs · 2025-05-01

**Decision:**

Accept (poster)

**Comment:**

This paper introduces Robust Multi-bit Text Watermarking using dual LLM-based paraphrasers and PPO-based co-training, achieving >99.99% AUC and high robustness under attacks. Reviewers' concerns on segmentation, efficiency, and adaptivity were addressed with strong ablations. The AC thus recommends acceptance of the paper. Human evaluation and broader reproducibility analysis remain areas for future improvement.